# Neutrophils in Extravascular Body Fluids: Cytological-Energy Analysis Enables Rapid, Reliable and Inexpensive Detection of Purulent Inflammation and Tissue Damage

**DOI:** 10.3390/life12020160

**Published:** 2022-01-21

**Authors:** Petr Kelbich, Petr Vachata, Vilem Maly, Tomas Novotny, Jan Spicka, Inka Matuchova, Tomas Radovnicky, Ivan Stanek, Jan Kubalik, Ondrej Karpjuk, Frantisek Smisko, Eva Hanuljakova, Jan Krejsek

**Affiliations:** 1Department of Biomedicine and Laboratory Diagnostics, Faculty of Health Studies, Jan Evangelista Purkinje University and Masaryk Hospital, 401 13 Usti nad Labem, Czech Republic; jan.spicka@kzcr.eu (J.S.); eva.hanuljakova@kzcr.eu (E.H.); 2Department of Clinical Immunology and Allergology, Faculty of Medicine and University Hospital, Charles University in Prague, 500 03 Hradec Kralove, Czech Republic; inka.matuchova@likvor.cz (I.M.); jan.kubalik@kzcr.eu (J.K.); jan.krejsek@fnhk.cz (J.K.); 3Laboratory for Cerebrospinal Fluid, Neuroimmunology, Pathology and Special Diagnostics Topelex, 190 00 Prague, Czech Republic; 4Department of Neurosurgery, Faculty of Health Studies, Jan Evangelista Purkinje University and Masaryk Hospital, 401 13 Usti nad Labem, Czech Republic; petr.vachata@kzcr.eu (P.V.); tomas.radovnicky@kzcr.eu (T.R.); 5Department of Neurosurgery, Faculty of Medicine and University Hospital, Charles University in Prague, 301 00 Pilsen, Czech Republic; 6Department of Thoracic Surgery, Masaryk Hospital, 401 13 Usti nad Labem, Czech Republic; vilem.maly@kzcr.eu (V.M.); ivan.stanek@kzcr.eu (I.S.); ondrej.karpjuk@kzcr.eu (O.K.); 7Department of Orthopaedics, Faculty of Health Studies, Jan Evangelista Purkinje University and Masaryk Hospital, 401 13 Usti nad Labem, Czech Republic; tomas.novotny@kzcr.eu; 8Department of Histology and Embryology, Second Faculty of Medicine, Charles University, 150 06 Prague, Czech Republic; 9Department of Orthopaedics, Regional Hospital, 360 01 Karlovy Vary, Czech Republic; frantisek.smisko@kkn.cz

**Keywords:** neutrophils, coefficient of energy balance, purulent inflammation, aspartate aminotransferase, cerebrospinal fluid, pleural effusion, abdominal effusion, synovial fluid

## Abstract

The simultaneous cytological and metabolic investigation of various extravascular body fluids (EBFs) provides clinically relevant information about the type and intensity of the immune response in particular organ systems. The oxidative burst of professional phagocytes with the concomitant production of reactive oxygen species consumes a large amount of oxygen and is the cause of switch to the development of anaerobic metabolism. We assessed the relationships between percentages of neutrophils, aerobic and anaerobic metabolism, and tissue damage via the determination of aspartate aminotransferase catalytic activities (AST) in cerebrospinal fluid (CSF), pleural effusions (PE), abdominal effusions (AE), and synovial fluids (SF). **EBFs with 0.0–20.0% neutrophils:** 83.0% aerobic and 1.3% strongly anaerobic cases with median of AST = 13.8 IU/L in CSF; 68.0% aerobic and 9.0% strongly anaerobic cases with median of AST = 20.4 IU/L in PE; 77.5% aerobic and 10.5% strongly anaerobic cases with median of AST = 18.0 IU/L in AE; 64.1% aerobic and 7.7% strongly anaerobic cases with median of AST = 13.8 IU/L in SF. **EBFs with 80.0–100.0% neutrophils:** 4.2% aerobic and 73.7% strongly anaerobic cases with median of AST = 19.2 IU/L in CSF; 7.4% aerobic and 77.3% strongly anaerobic cases with median of AST = 145.2 IU/L in PE; 11.8% aerobic and 73.7% strongly anaerobic cases with median of AST = 61.8 IU/L in AE; 25.5% aerobic and 38.2% strongly anaerobic cases with median of AST = 37.2 IU/L in SF. The significant presence of neutrophils, concomitant strong anaerobic metabolism, and elevated AST in various EBFs are reliable signs of damaging purulent inflammation.

## 1. Introduction

The detection of purulent inflammation is clinically essential to investigate various extravascular body fluids (EBFs). Attention is usually paid to the presence of neutrophils, glucose and lactate concentrations, and microbial analysis, either individually or simultaneously. Many authors describe the predominance of neutrophils, low glucose concentrations, high lactate concentrations, and the detection of extracellular bacteria in EBF as typical laboratory evidence of purulent inflammation [1,2,3,4,5,6,7,8,9,10,11,12,13,14]. The ability of each of these parameters to provide reliable information is still limited. The presence of neutrophils in EBF alone is not evidence of purulent inflammation in many patients. Similarly, changes in glucose and lactate concentrations in EBF may not provide precise information about local energy status. The concentration of glucose in EBF is dependent on the concentration of glucose in the blood. Hypoglycemia may mimic the lack of glucose in the examined compartment. However, hyperglycemia can lead to increased concentrations of glucose in EBF and can mask glucose consumption during inflammation in the affected body compartment [15,16]. Furthermore, the generally accepted scheme of glucose metabolism indicates that the concentration of lactate in EBF is not influenced exclusively by the extent of anaerobic metabolism in EBF but also by the supply of the energy substrate, i.e., glucose (Figure 1) [17,18,19].

Neutrophils are the principal cells of innate immunity. They serve as the first line of defense against predominantly bacterial invasion or exposure to exogenous dangerous materials [20,21]. Neutrophils are cells specialized in phagocytosis, and play a significant role at the beginning of inflammation and control the subsequent development of the inflammatory response. The immunobiological activities of these cells are numerous. Their primary role is to internalize either invading microbes or foreign particles or trap them through the formation of neutrophil extracellular traps (NETs). Internalization is followed by activation of the NADPH enzymatic complex and the production of reactive oxygen species (ROS). In addition, there is a release of granule microbicidal proteins, which are responsible for oxygen-independent mechanisms of microbial killing [22,23,24,25,26,27,28,29,30,31,32,33]. ROS production, i.e., oxidative burst, is a powerful antimicrobial weapon, and a major component of the innate immune defense against bacterial and fungal infections [22,26,27,30].

We consider the cytological-energy analysis of EBFs a reliable approach to detect local inflammatory response in the relevant organ system. This method also provides information about its type and intensity. The first step is the cytological analysis. The cellular composition of EBF alone can be insufficient to determine the type of inflammation. It has to be supplemented by data evidencing its metabolic activities. Therefore, knowledge about the activity of the presented immunocompetent cells is necessary. We have recognized, as a reliable functional parameter, the so-called coefficient of energy balance (KEB) (1) [15,16,34,35,36,37]:(1)$$\mathrm{KEB}=38-18\frac{\left[\mathrm{lactate}\right]}{\left[\mathrm{glucose}\right]}$$

Legend:

[glucose] = molar concentration of glucose in the EBF (mMol/L)

[lactate] = molar concentration of lactate in the EBF (mMol/L).

The KEB is defined as the theoretical average number of adenosine triphosphate molecules produced from one molecule of glucose under the conditions found in the relevant extravascular compartment. High KEB values (over 28.0) represent aerobic metabolism in EBF. KEB values between 20.0 and 28.0 represent slight anaerobic metabolism in EBF. KEB values between 10.0 and 20.0 represent moderate anaerobic metabolism in EBF. KEB values under 10.0 represent strong anaerobic metabolism [15,16,34,35,36,37].

The oxidative burst of neutrophils with the production of ROS consumes large amounts of oxygen [21,30,38,39,40]. The enzyme responsible for the oxygen consumption is an NADPH oxidase complex [21,30]. It is the hallmark of purulent inflammation. Its consequence is a fast switch to anaerobic metabolism in EBF presented by decreasing KEB values. Our previous studies prove that the significant presence of neutrophils in EBF with KEB values below 10.0 is reliable evidence of local purulent inflammation in the corresponding EBF [15,16,34,35,36,37].

The oxidative burst of neutrophils in purulent inflammation is very aggressive and often harmful to host cells and tissue [24,27,31,39,40,41,42]. Therefore, a significant part of cytological-energy analysis is also to detect tissue damage using the catalytic activity of aspartate aminotransferase (AST) in EBF [36,43]. AST is a non-specific enzyme. Therefore, it can be released from various cells, including blood cells. We are able to avoid the influence of erythrocytes by excluding hemolytic samples. On the other hand, it is important to interpret the results of laboratory analysis with respect to the proportion of AST released directly from dying neutrophils [44].

## 2. Material and Methods

This retrospective study was approved by the local Ethics Committee of the Masaryk Hospital Usti nad Labem, Czech Republic (reference number: 301/15). No informed consent was required for this study. The work did not involve any human experiments and did not require the collection of data outside of routine parameters. All patient records and information were anonymized and deidentified.

We carried out the cytological-energy analysis of 4268 samples of cerebrospinal fluids, 2668 samples of pleural effusions, 445 samples of abdominal effusions, and 211 samples of synovial fluids taken from patients suffering from different pathologies. Our aim was to compare the significance of neutrophils in compartments in which these extravascular body fluids are formed. We focused on the number of cells in EBF, cellular composition, local energy metabolism using the coefficient of energy balance (KEB), and the catalytic activity of aspartate aminotransferase (AST) [15,16,34,35,36,37,43].

### 2.1. Cytological-Energy Analysis of Extravascular Body Fluids

The samples of cerebrospinal fluid, pleural effusion, abdominal effusion, and synovial fluids were collected in the standard way via test tubes without anticoagulants and immediately transported to our clinical laboratory. The total number of elements in these extravascular body fluids was counted under the optical microscope using a Fuchs–Rosenthal chamber, and microscopic smear using a cytocentrifuge method was prepared immediately after receiving the sample in all cases. Permanent cytological smears were stained using Hemacolor (Merck Co., Gernsheim, Germany). Microscopic analyses were performed by trained laboratory personal using an Olympus BX40 microscope (Olympus, Japan) to determine cellular composition of EBFs.

Another aliquot of the samples was centrifuged and the molar concentration of glucose using the hexokinase method, lactate using the lactate oxidase and peroxidase method, and catalytic activities of aspartate aminotransferase (AST) using the IFCC method on a Cobas 6000 analyzer (Roche Diagnostics, Basel, Switzerland) were determined.

KEB values were calculated for all samples, including rare cases with very low glucose concentrations below the measurement limit (=0.11 mmol/L). A glucose concentration of 0.11 mmol/L was then used as the minimum for all these deeply anaerobic cases.

### 2.2. Statistical Analysis

The individual groups of investigated samples were divided into subgroups of 20% of neutrophils. All evaluated parameters are presented in tables as the median and the 1st and 3rd interquartile range.

Statistical analysis was performed using the ANOVA Kruskal–Wallis test. All statistical tests were performed using Statistica 14.0 software (StatSoft Inc., Tulsa, OK, USA). *p* values < 0.05 were considered as significant.

## 3. Results

Table 1, Table 2, Table 3 and Table 4 show the increasing number of nucleated cells along with the increasing percentage of neutrophils in all analyzed EBFs. The most significant statistical differences are seen in pleural and abdominal effusions, they are less pronounced in cerebrospinal fluid, and even less in synovial fluid.

The concomitant increase in anaerobic metabolism along with an increase in neutrophil percentage in cerebrospinal fluid, pleural effusions, abdominal effusions, and synovial fluid was observed. The most significant statistical differences were found in pleural effusions, they were less pronounced in abdominal effusions, even less in cerebrospinal fluid, and the least in synovial fluid (Table 1, Table 2, Table 3 and Table 4).

Finally, we found the increase in AST catalytic activity along with the increase in neutrophil percentage as the most statistically significant in pleural effusions, as less significant in abdominal fluids, and as the least significant in cerebrospinal fluid and synovial fluid (Table 1, Table 2, Table 3 and Table 4).

Figure 2, Figure 3, Figure 4 and Figure 5 show the distribution of cases with aerobic (KEB > 28.0), slightly anaerobic (KEB = 20.0–28.0), moderately anaerobic (KEB = 10.0–20.0), and strongly anaerobic (KEB < 10.0) EBFs in correlation with percentages of neutrophils. Figure 2, Figure 3 and Figure 4 are very similar. The frequencies of cases with aerobic metabolism decrease and the frequencies of cases with strong anaerobic metabolism increase with increased relative presence of neutrophils in cerebrospinal fluid, pleural effusions, and abdominal effusions. We also observed a decrease in aerobic cases along with an increase in neutrophil percentage in synovial fluids. Despite this phenomenon, the simultaneous increase in strongly anaerobic cases is relatively small (Figure 5). While the frequency of strongly anaerobic cases in cerebrospinal fluid, pleural effusions, and abdominal effusions reaches more than 70.0%, it is only 38.2% in synovial fluids.

Figure 6, Figure 7, Figure 8, Figure 9, Figure 10, Figure 11, Figure 12 and Figure 13 compare the energy ratios in cerebrospinal fluid, pleural effusions, abdominal effusions, and synovial fluid between reactive neutrophil-dominated states and purulent inflammation.

## 4. Discussion

Neutrophils are the major phagocytic cells and the final effector cells of innate immunity, with a primary role in the clearance of extracellular pathogens [29]. Their predominant presence in extravascular body fluids (EBFs) is usually recognized as a purulent inflammatory response induced by extracellular bacteria. This could sometimes be true but sometimes not. The presence of neutrophils in the relevant area generally only informs about the mobilization of innate immunity, but not about the nature of these responses. Information about the metabolic activity of these cells is essential to conclude. Therefore, we measured the molar concentrations of glucose and lactate in EBFs and calculated the coefficient of energy balance (KEB) [15,16,34,35,36,37,43]. The rationale of this approach is based on the parallel increase in local inflammation intensity and oxygen utilization, and the development of anaerobic metabolism. Especially, an oxidative burst of professional phagocytes, including neutrophils with production of ROS, is the cause of large amounts of oxygen consumption [21,25,30,38,39,40,45]. Therefore, strong anaerobic metabolism enables us to distinguish between a purulent inflammatory response and a non-purulent one (Figure 6, Figure 7, Figure 8, Figure 9, Figure 10, Figure 11, Figure 12 and Figure 13). Our recent studies revealed low KEB values (<10.0) as a reliable marker of neutrophils oxidative burst in the term of purulent inflammation [15,16,34,35,36,37].

We compared the development of cytological-energy parameters in cerebrospinal fluid, pleural effusions, abdominal effusions, and synovial fluid. A simultaneously increased number of nucleated cells together with an increased percentage of neutrophils and increased anaerobic metabolism is apparent in all types of EBFs. This pattern is almost the same when compared with the cerebrospinal fluid, pleural effusions, and abdominal effusions. We found that more than 70.0% of cases with very high percentages of neutrophils (80.0%–100.0%) are associated with strong anaerobic metabolism (KEB < 10.0) (Table 1, Table 2 and Table 3; Figure 2, Figure 3 and Figure 4). This trend is also evident in the synovial fluid. Nevertheless, only 38.2% of cases fully express the signs of purulent inflammation. Frequently, cases with high percentages of neutrophils were associated with high numbers of aerobic, slightly anaerobic, and moderately anaerobic EBFs (Table 4 and Figure 5). We call this condition “preventive protection” against bacterial infection, or exposure to exogenous dangerous materials and simultaneously increased risk of purulent inflammation. It is possible to classify these conditions into five classes (1 to 5) reflecting the degree of risk [16]. Grade 1 represents no risk, grade 2 represents a moderate risk, grade 3 represents a significant risk, grade 4 represents a high risk, and grade 5 represents purulent inflammation. This approach is now adopted into our laboratory and clinical practice to prevent the development of purulent inflammation.

An integral part of our EBF analysis is the determination of AST catalytic activity. This ubiquitous enzyme is released from damaged cells into the extracellular space. Therefore, increased values of AST catalytic activity in EBFs provide information about cellular and tissue damage [43]. The simultaneous increase in cell number, relative neutrophil count, anaerobic metabolism, and catalytic activity of AST in EBFs heralds increased cellular and tissue damage along with increased intensity of inflammation. The most severe cellular and tissue damage was found in all purulent EBFs (Table 1, Table 2, Table 3 and Table 4; Figure 7, Figure 9, Figure 11 and Figure 13). This is in accordance with the results of many authors that the oxidative burst of neutrophils in purulent inflammation is very aggressive, and often associated with tissue damage and the amount of dying neutrophils [24,27,31,39,40,41,42,44].

## 5. Conclusions

Neutrophils are the first innate immune cells that are rapidly recruited from the bloodstream to sites of infection or tissue damage. We are convinced that it is important both for laboratory and clinical practice to distinguish between the non-purulent activity of neutrophils and aggressive purulent inflammation in different extracellular body fluids (EBFs). In accordance with our long-term experience, the solution is cytological-energy analysis, which enables their energy ratios to be revealed. We found the concomitant increase in nucleated cells (especially leukocytes), percentages of neutrophils, and anaerobic metabolism using the KEB values, and structural tissue damage using AST catalytic activities in the cerebrospinal fluid, pleural effusions, abdominal effusions, and the synovial fluid. The strong anaerobic metabolism in the relevant EBF (KEB < 10.0) with the significant occurrence of neutrophils represents reliable evidence of their oxidative burst in the term of purulent inflammation. This inflammatory response is usually accompanied by significant cellular and tissue destruction.

The relationships between the intensity of inflammation and tissue damage are very similar in all EBFs investigated by us. Nevertheless, their values fluctuate. Neutrophils are vital for the defense of an organism. On the other hand, they could be harmful to the organism itself. The interpretation of the presence of neutrophils in EBFs taken from different body compartments has to be cautious. The smallest presence of neutrophils in the CSF represents the highest protection of the CNS against damage caused by inflammatory response. Higher neutrophil counts in pleural and abdominal effusions reflect a lower degree of pleural and abdominal protection. The highest number of neutrophils in the synovial fluids indicates the lowest protection of the periphery from purulent inflammation.

## Figures and Tables

**Figure 1 life-12-00160-f001:**
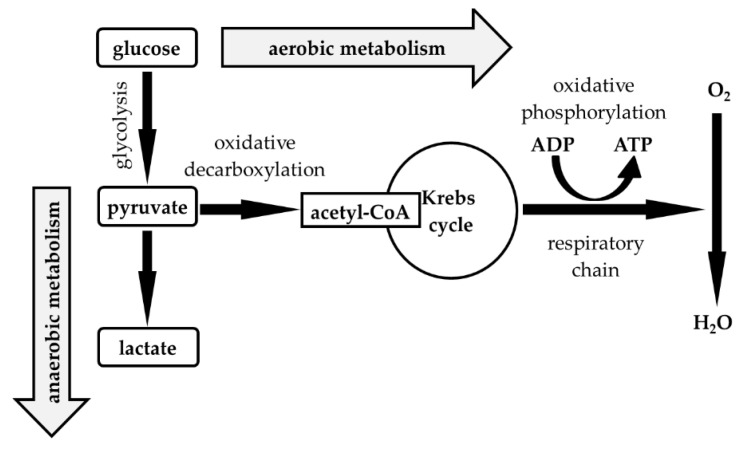
Energy metabolism of glucose. Legend: acetyl-CoA—acetyl coenzyme A; ADP—adenosine diphosphate; ATP—adenosine triphosphate; O_2_—oxygen; H_2_O—water.

**Figure 2 life-12-00160-f002:**
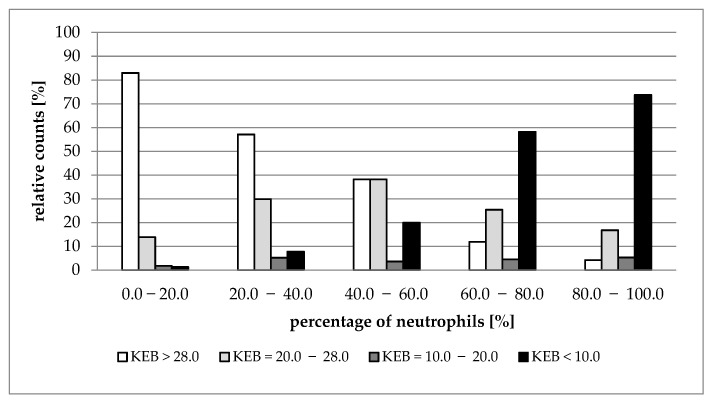
Distribution of KEB values according to the percentage of neutrophils in cerebrospinal fluid.

**Figure 3 life-12-00160-f003:**
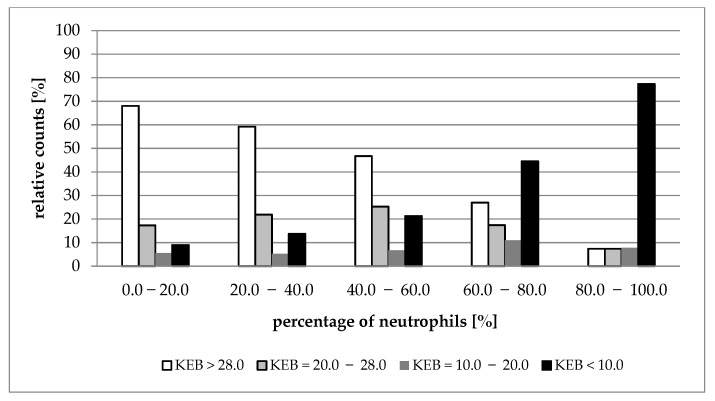
Distribution of KEB values according to the percentage of neutrophils in pleural effusions.

**Figure 4 life-12-00160-f004:**
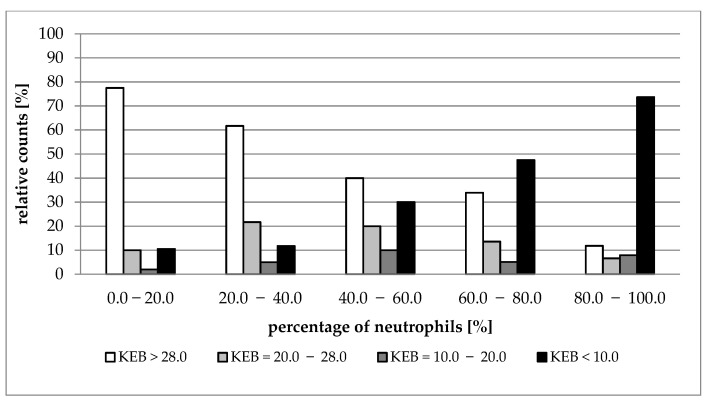
Distribution of KEB values according to the percentage of neutrophils in abdominal effusions.

**Figure 5 life-12-00160-f005:**
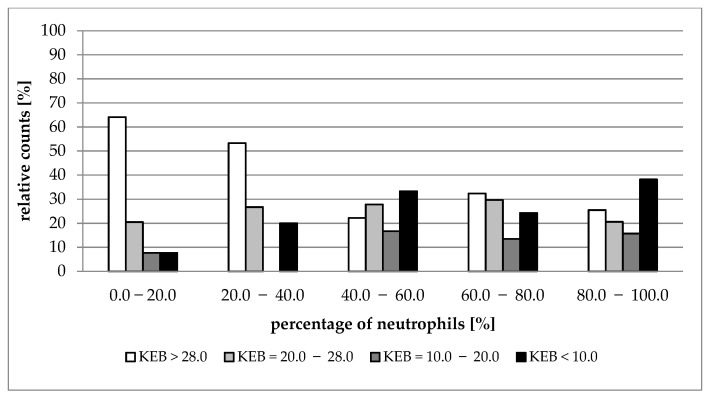
Distribution of KEB values according to the percentage of neutrophils in synovial fluids.

**Figure 6 life-12-00160-f006:**
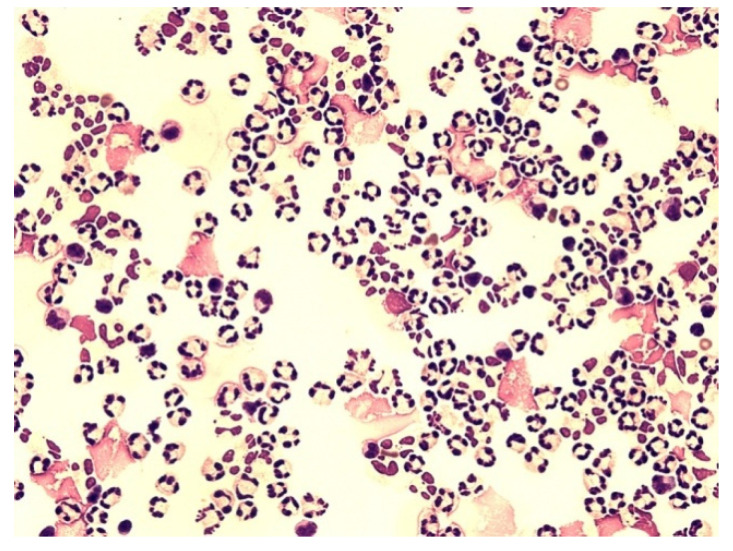
Predominance of neutrophils in the cerebrospinal fluid; KEB = 22.9; non-purulent inflammatory reaction in the CNS of patients caused by subarachnoid hemorrhage.

**Figure 7 life-12-00160-f007:**
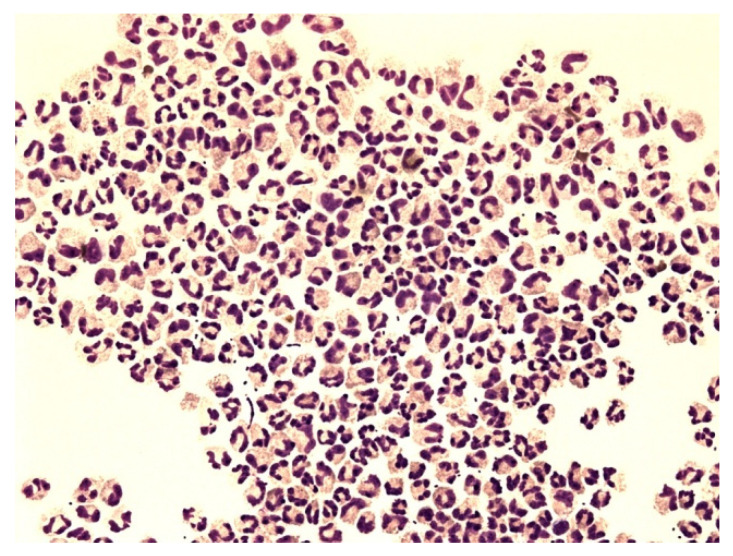
Predominance of neutrophils and bacteria in the cerebrospinal fluid; KEB = −4743.5; purulent inflammation in the CNS induced by infection *Streptococcus pneumoniae*.

**Figure 8 life-12-00160-f008:**
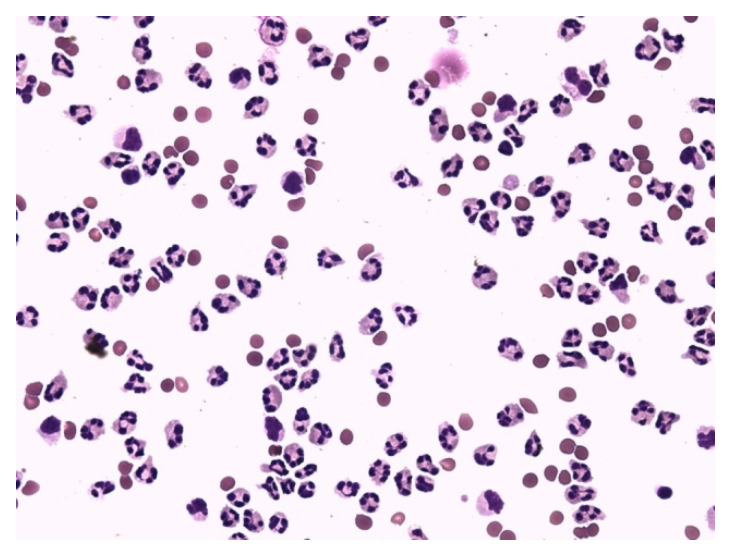
Predominance of neutrophils in pleural effusion; KEB = 32.5; non-purulent reaction in the pleural cavity of patients with atopic bronchial asthma.

**Figure 9 life-12-00160-f009:**
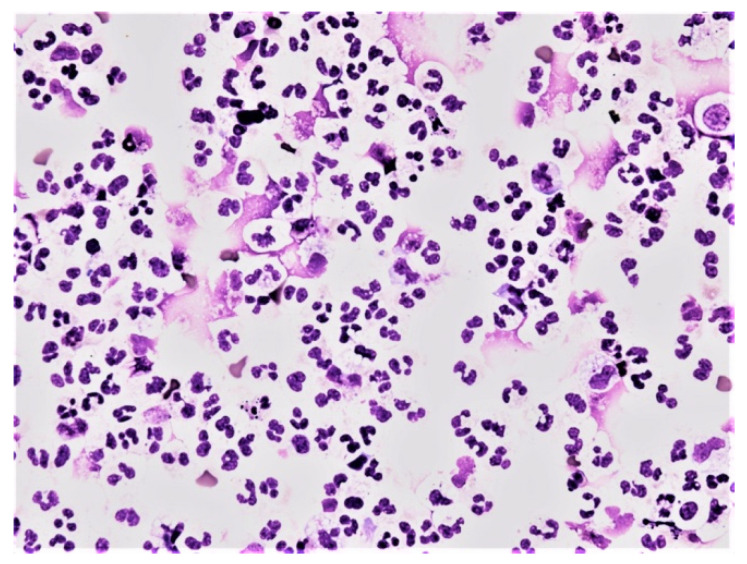
Predominance of neutrophils in pleural effusion; KEB = −4015.4; purulent inflammation in the pleural cavity induced by *Streptococcus constellatus*.

**Figure 10 life-12-00160-f010:**
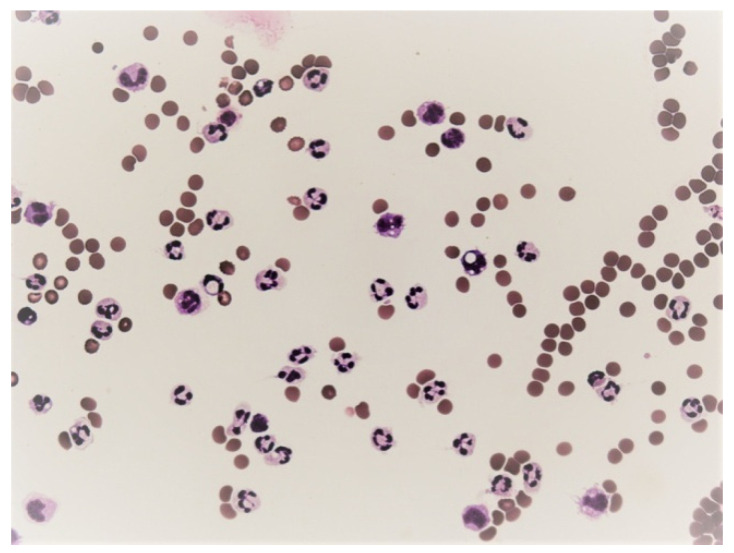
Predominance of neutrophils in abdominal fluid; KEB = 34.7; non-purulent reaction in the abdominal cavity of patients one day after the robotic radical prostatectomy.

**Figure 11 life-12-00160-f011:**
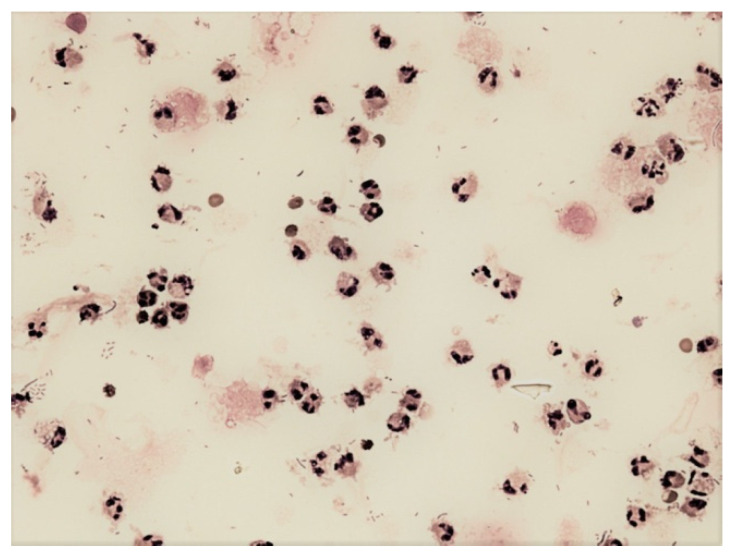
Predominance of neutrophils and bacteria in abdominal fluid; KEB = −1434.7; purulent inflammation of bacterial etiology in the abdominal cavity.

**Figure 12 life-12-00160-f012:**
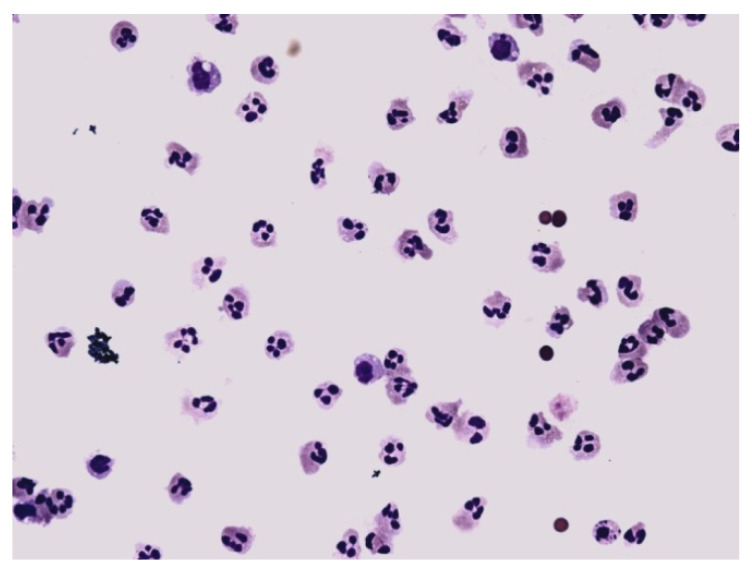
Predominance of neutrophils in synovial fluid; KEB = 32.7; non-purulent response in the knee joint after traumatic distortion.

**Figure 13 life-12-00160-f013:**
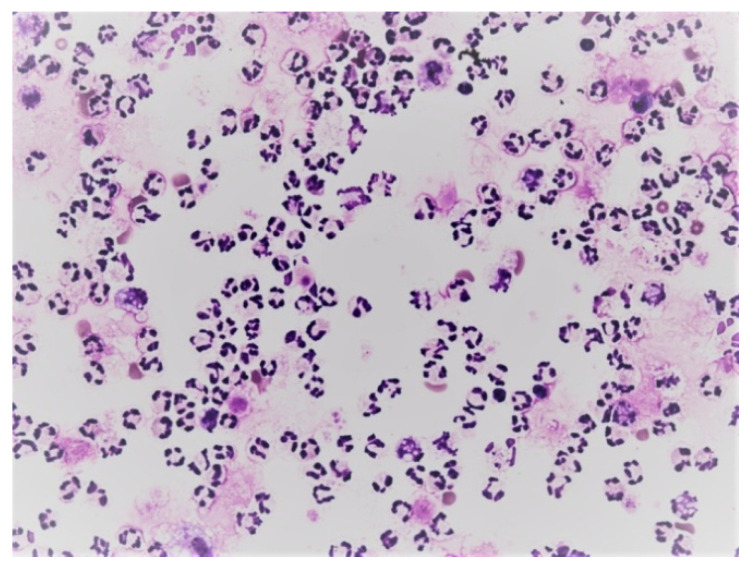
Predominance of neutrophils in synovial fluid; KEB = −855.5; purulent inflammation in the knee joint induced by the presence of *Pseudomonas aeruginosa*.

**Table 1 life-12-00160-t001:** Cytological-energy analysis of cerebrospinal fluid. Legend: Groups sharing capital letters (A, B, C) are not significantly different as evidenced by ANOVA Kruskal–Wallis test for multiple comparisons (family wise α = 0.05); A/B—no significant difference with both groups “A” and “B”; IQR—interquartile range; KEB—coefficient of energy balance; AST—aspartate aminotransferase catalytic activity. B/C—no significant difference with both groups “B” and “C”.

Median (1st–3rd IQR) Neutrophils (%)	0.0–20.0	20.0–40.0	40.0–60.0	60.0–80.0	80.0–100.0
Number of Patients	3974	77	55	67	95
Nucleated cells (elements/1 µL)	A **2** (1–4)	B **12** (1–117)	B/C **48** (7–261)	C **421** (72–1195)	C **2304** (496–6059)
KEB	A **29.83** (28.76–30.67)	B **28.29** (25.59–30.00)	B **26.45** (20.73–29.78)	C **−3.33** (−928.93–24.97)	C **−62.87** (−1099.94–14.42)
AST (IU/L)	A **13.8** (10.8–17.4)	A **12.0** (10.2–17.4)	A/B **15.0** (10.8–18.0)	B **18.6** (12.6–25.8)	B **19.2** (13.2–40.8)

**Table 2 life-12-00160-t002:** Cytological-energy analysis of pleural effusions. Legend: Groups sharing capital letters (A, B, C, D, E) are not significantly different as evidenced by ANOVA Kruskal–Wallis test for multiple comparisons (family wise α = 0.05); IQR—interquartile range; KEB—coefficient of energy balance; AST—aspartate aminotransferase catalytic activity.

Median (1st–3rd IQR) Neutrophils (%)	0.0–20.0	20.0–40.0	40.0–60.0	60.0–80.0	80.0–100.0
Number of Patients	1242	306	225	344	551
Nucleated cells (elements/1 µL)	A **683** (268–1707)	A **681** (224–1707)	B **960** (344–2453)	C **1840** (566–5120)	D **7307** (1559–34,987)
KEB	A **31.13** (26.03–33.60)	B **29.48** (24.0–33.40)	C **27.53** (17.40–31.90)	D **15.22** (−86.99–28.19)	E **−166.61** (−2202.18–5.56)
AST (IU/L)	A **20.4** (12.6–33.6)	B **25.8** (15.0–46.8)	C **34.2** (16.8–73.8)	D **59.4** (31.2–159.0)	E **145.2** (65.4–393.6)

**Table 3 life-12-00160-t003:** Cytological-energy analysis of abdominal effusions. Legend: Groups sharing capital letters (A, B, C, D) are not significantly different as evidenced by ANOVA Kruskal–Wallis test for multiple comparisons (family wise α = 0.05); A/B—no significant difference with both groups “A” and “B”; B/C—no significant difference with both groups “B” and “C”; C/D—no significant difference with both groups “C” and “D”; IQR—interquartile range; KEB—coefficient of energy balance; AST—aspartate aminotransferase catalytic activity.

Median (1st–3rd IQR) Neutrophils (%)	0.0–20.0	20.0–40.0	40.0–60.0	60.0–80.0	80.0–100.0
Number of Patients	200	60	50	59	76
Nucleated cells (elements/1 µL)	A **200** (30–536)	A/B **309** (160–965)	B/C **1275** (300–3051)	C/D **1621** (504–7707)	D **4693** (1596–23,520)
KEB	A **31.59** (28.71–33.68)	A/B **30.40** (23.31–32.34)	B/C **25.27** (4.55–31.32)	C/D **17.62** (−123.21–29.86)	D **−38.71** (−503.78–15.42)
AST (IU/L)	A **18.0** (12.0–27.0)	A/B **20.4** (15.6–28.8)	B/C **27.0** (15.0–67.2)	C/D **34.8** (17.4–95.4)	D **61.8** (27.0–161.4)

**Table 4 life-12-00160-t004:** Cytological-energy analysis of synovial fluids. Legend: Groups sharing capital letters (A, B) are not significantly different as evidenced by ANOVA Kruskal–Wallis test for multiple comparisons (family wise α = 0.05); A/B—no significant difference with both groups “A” and “B”; IQR—interquartile range; KEB—coefficient of energy balance; AST—aspartate aminotransferase catalytic activity.

Median (1st–3rd IQR) Neutrophils (%)	0.0–20.0	20.0–40.0	40.0–60.0	60.0–80.0	80.0–100.0
Number of Patients	39	15	18	37	102
Nucleated cells (elements/1 µL)	A **187** (89–867)	A **1451** (439–3413)	A/B **4267** (2133–6667)	B **6547** (3539–32,171)	B **20,693** (6827–53,760)
KEB	A **30.50** (25.17–32.37)	A/B **28.17** (20.74–29.28)	B **19.88** (5.84–27.57)	B **26.80** (10.46–29.12)	B **18.79** (−9.18–28.12)
AST (IU/L)	A **13.8** (12.0–19.8)	B **24.0** (21.0–33.0)	B **27.6** (17.4–31.8)	B **28.2** (19.2–55.8)	B **37.2** (25.2–64.2)

## Data Availability

All relevant data are within the paper.

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
