# Peer review of "Neutrophils in Extravascular Body Fluids: Cytological-Energy Analysis Enables Rapid, Reliable and Inexpensive Detection of Purulent Inflammation and Tissue Damage"

_life, 2022, doi:10.3390/life12020160_

Round 1
Reviewer 1 Report
The manuscript entitled “Neutrophils in extravascular body fluids: cytological-energy analysis enables rapid, reliable and inexpensive detection of purulent inflammation and tissue damage” deals with the study of the detection of purulent inflammation in various extravascular body fluids, namely, cerebrospinal fluid, pleural effusion, abdominal effusion, and synovial fluid. To evaluate the purulent inflammation, the authors considered the number of cells, the coefficient of energy balance (KEB), based on glucose and lactate concentrations, and the catalytic activity of aspartate aminotransferase. The authors retrospectively analysed 4268 samples of cerebrospinal fluids, 2668 samples of pleural effusions, 445 samples of abdominal effusions and 211 samples of synovial fluids, dividing in the patients in four strata, according to their KEB values.
The manuscript is well written and easy to understand. The experiments carried out were enough and suitable for the purpose of the manuscript. The references used in the manuscript are recent and adequate. Regarding the novelty of the manuscript, although not being the first incursion of the authors exploring the detection of purulent inflammation, as far as I am concerned this is the first time that is studied in these four extravascular fluids using these parameters.
In my opinion, the results shown in this manuscript are interesting for a broader community and deserve to be published, nonetheless. Despite its great potential, the manuscript comes with some small issues that need to be addressed:
- The authors should revise the bibliography, it lacks the number of the references.
- Figures 2 – 5, in the abscissae axis I would recommend the authors to change the name to percentage of neutrophils [%] for the sake of consistency.
- Figures 6 – 13, as they show results, I would advise the authors to relocate said figures in the results section.
- Line 134, change activities for activity.
- Line 163, Change Biureth for Biuret method, and include a reference for the method used.
- Line 166, Separate the degree from the number when expressing temperatures.
- Line 172, write the p value in italics.
Best regards
Author Response
Dear reviewer,
Thank you very much for your positive assessment of our work and for recommendations which will improve our text.
Here is the list of your recommendations and our solutions (the new text in the manuscript is written in red):
- The authors should revise the bibliography, it lacks the number of the references.
It was corrected.
- Figures 2 – 5, in the abscissae axis I would recommend the authors to change the name to percentage of neutrophils [%] for the sake of consistency.
It was done.
- Figures 6 – 13, as they show results, I would advise the authors to relocate said figures in the results section.
It was done.
- Line 134, change activities for activity.
It was corrected.
- Line 163, Change Biureth for Biuret method, and include a reference for the method used.
We did not evaluate this parameter in our study. Therefore I completely excluded this redundant information.
- Line 166, Separate the degree from the number when expressing temperatures.
I have discarded the information on sample storage as unnecessary.
- Line 172, write the p value in italics.
It was done.
Sincerely yours,
Petr Kelbich et al.

Reviewer 2 Report
The Authors present results of an impressively large series of samples of various extravascular body fluids (EBFs). They compare results of several tests that enable reliable detection of purulent inflammation when used in combination (cell count; neutrophil percentage; glucose and lactate with calculation of so-called coefficient of energy balance /KEB/ invented by the Authors previously). Panel of tests recommended is highly relevant for routine practice and comparison among various test results is very interesting. I appreciate the Figures 6-13 providing the reader with relevant examples.
The major concern of the reviewer is the interpretation of elevated AST activities as signs of tissue destruction. Whereas the previous outstanding study of the Authors´ group (Kelbich et al. Brain Sci 2020;10:698) on patients after CNS haemorrhage makes this interpretation very plausible in case of CSFs, possible sources of elevated AST values in other EBFs should be more thoroughly discussed. Namely, neutrophils contain AST too (Curi et al. Braz J Med Biol Res 1999; 32(1): 15-21, Table 1 on Page 16), and correlation between the neutrophil proportion, cell count and elevated AST values might be significantly related to the release of AST from dying neutrophils themselves in case of purulent inflammation. Hence, some doubt may be cast upon the Authors´ claim attributing elevated AST values to tissue damage. At least, possible causes of elevated AST values should be discussed in more detail.
Specific comments:
INTRODUCTION: Well written. On Page 3, row 93, ‘relevant organ system’ is perhaps more appropriate than ‘relevant organic system’. Legend to the KEB formula should containe glucose and lactate concentration in the EBF and not in the CSF only.
MATERIALS AND METHODS: Patient stratification according to KEB values was presented by the Authors previously (Ref. 16); in the text, point 3) (KEB 20.0 – 10.0) is very difficult to understand and should be rewritten. Perhaps this paragraph could be presented in a more condensed form or even deleted since the basic information is provided already in the Introduction section.
Page 4, row 160: … cellular composition of EBFs (rather than pleural effusions only).
Page 4, row 163 ‘biuret’ instead of ‘Biureth’
Statistical analysis: Is the description 1st and 3rd quartile equivalent to a more common term Interquartal range (IQR)?
RESULTS: Tables 1-4 are very instructive. Nevertheless, the meaning of A/B, B/C, C/D in terms of statistical differences should be explained.
Page 8, row 222: ‘… the frequency of strongly anaerobic in cerebrospinal … ’ – I believe there is a missing word in the sentence. Should it be read ‘… the frequency of strongly anaerobic cases in cerebrospinal …’?
Figure 6: ‘… patients cost by subarachnoid haemorrhage’: did the Authors mean ‘caused’?
Figure Legends: I believe KEB values should have no more than one decimal point.
DISCUSSION: Page 11, row 266: The term ‘dynamics’ usually refers to a temporal change that has not been investigated in this study. Perhaps the term ‘various cytological-energy parameters’ would be more appropriate.
Page 11, row 271: 70.0% of cases
Page 11, row 277: ‘characterized by increased risk’ rather than ‘characterize’
Page 11, row 278 ff. This is quite different from the Authors´ previous definition of preventive protection (neutrophil predominance AND KEB 15 or greater in Ref. 16). The Authors should explain this more clearly. If I understand it correctly, classes 1 to 5 determine the risk of purulent inflammation and not the “preventive protection” itself.
Page 11, last paragraph: As proposed earlier in these Comments, the cause of increased AST activity should be discussed more cautiously. In my opinion, there is insufficient evidence – at least beyond the CSF compartment - to attribute it solely to ‘tissue damage’ as proposed by the Authors.
CONCLUSIONS: Page 12, row 298: ‘In accordance’ rather than ‘In according’
Page 12, row 311: ‘compartments’ rather than ‘departments’
The last paragraph of the Conclusions section is again difficult to understand and should be re-written.
LITERATURE: References are not numbered (perhaps numbers were lost when adding row numbers).
Final remark (optional): The concept of the coefficient of energy balance could possibly also be improved to correspond more closely to the Authors´ definition (Refs. 15 and 16 and this manuscript, Page 3, rows 105-107) if the Authors could find an appropriate correction of the formula to prevent negative values (according to the definition provided here and in the Authors´ earlier study /Ref. 15/, the result should be between 2 and 38). The second reason is that KEB in the present form cannot be calculated in cases with unmeasurably low glucose. Such cases are not uncommon (I suppose value of 0.1 mmol/L was used by the Authors in such cases instead). The Authors might wish to discuss this issue in the Discussion section. Possible relevance of adding pyruvate determination in further studies could be discussed as well; namely pyruvate is the metabolite standing on the intersection of different metabolic pathways as showed in Figure 1.
Author Response
Dear reviewer,
Thank you very much for your positive assessment of our work and for recommendations which will improve our text.
Here is the list of your recommendations and our solutions (the new text in the manuscript is written in red):
- The major concern of the reviewer is the interpretation of elevated AST activities as signs of tissue destruction. Whereas the previous outstanding study of the Authors´ group (Kelbich et al. Brain Sci 2020;10:698) on patients after CNS haemorrhage makes this interpretation very plausible in case of CSFs, possible sources of elevated AST values in other EBFs should be more thoroughly discussed. Namely, neutrophils contain AST too (Curi et al. Braz J Med Biol Res 1999; 32(1): 15-21, Table 1 on Page 16), and correlation between the neutrophil proportion, cell count and elevated AST values might be significantly related to the release of AST from dying neutrophils themselves in case of purulent inflammation. Hence, some doubt may be cast upon the Authors´ claim attributing elevated AST values to tissue damage. At least, possible causes of elevated AST values should be discussed in more detail.
I agree. I added notice about this issue into the “Introduction”.
Specific comments:
- INTRODUCTION: Well written. On Page 3, row 93, ‘relevant organ system’ is perhaps more appropriate than ‘relevant organic system’. Legend to the KEB formula should containe glucose and lactate concentration in the EBF and not in the CSF only.
It has been corrected.
- MATERIALS AND METHODS: Patient stratification according to KEB values was presented by the Authors previously (Ref. 16); in the text, point 3) (KEB 20.0 – 10.0) is very difficult to understand and should be rewritten. Perhaps this paragraph could be presented in a more condensed form or even deleted since the basic information is provided already in the Introduction section.
I agree. It was useless to present the repeated information. Therefore I deleted the relevant part of the text.
- Page 4, row 160: … cellular composition of EBFs (rather than pleural effusions only).
It has been corrected.
- Page 4, row 163 ‘biuret’ instead of ‘Biureth’
We did not asses total protein concentrations in our study. Therefore I deleted this redundant information about “biuret”.
- Statistical analysis: Is the description 1st and 3rd quartile equivalent to a more common term Interquartal range (IQR)?
I usually use the term “1st and 3rd quartile”. The term “IQR” is absolutely acceptable for me. Therefore I have rewritten it.
- RESULTS: Tables 1-4 are very instructive. Nevertheless, the meaning of A/B, B/C, C/D in terms of statistical differences should be explained.
I have added explanation into legends.
- Page 8, row 222: ‘… the frequency of strongly anaerobic in cerebrospinal … ’ – I believe there is a missing word in the sentence. Should it be read ‘… the frequency of strongly anaerobic cases in cerebrospinal …’?
I added “cases” into the text.
- Figure 6: ‘… patients cost by subarachnoid haemorrhage’: did the Authors mean ‘caused’?
It has been corrected.
- Figure Legends: I believe KEB values should have no more than one decimal point.
I traditionally use two decimal points in my practice. Only one decimal point is acceptable. I have corrected it.
- DISCUSSION: Page 11, row 266: The term ‘dynamics’ usually refers to a temporal change that has not been investigated in this study. Perhaps the term ‘various cytological-energy parameters’ would be more appropriate.
Yes, I agree. I have re-written “determined dynamics” to “compared development”.
- Page 11, row 271: 70.0% of cases
It has been corrected.
- Page 11, row 277: ‘characterized by increased risk’ rather than ‘characterize’
It has been corrected.
- Page 11, row 278 ff. This is quite different from the Authors´ previous definition of preventive protection (neutrophil predominance AND KEB 15 or greater in Ref. 16). The Authors should explain this more clearly. If I understand it correctly, classes 1 to 5 determine the risk of purulent inflammation and not the “preventive protection” itself.
I understand, that the link of the terms “preventive protection” and “risk of purulent inflammation” seems to be controversially. I traditionally use it in my medical practice because my aim is to express together necessity of these cells for defense of organism against different danger factors and their own serious danger for organism at the same time. I tried to make the relevant sentence in the text more accuracy.
- Page 11, last paragraph: As proposed earlier in these Comments, the cause of increased AST activity should be discussed more cautiously. In my opinion, there is insufficient evidence – at least beyond the CSF compartment - to attribute it solely to ‘tissue damage’ as proposed by the Authors.
I have added mention about participation of dying neutrophils.
- CONCLUSIONS: Page 12, row 298: ‘In accordance’ rather than ‘In according’
It has been corrected.
- Page 12, row 311: ‘compartments’ rather than ‘departments’
It has been corrected.
- The last paragraph of the Conclusions section is again difficult to understand and should be re-written.
I have re-written the particular text. I hope it’s clearer.
- LITERATURE: References are not numbered (perhaps numbers were lost when adding row numbers).
It has been corrected.
Final remark (optional):
- The concept of the coefficient of energy balance could possibly also be improved to correspond more closely to the Authors´ definition (Refs. 15 and 16 and this manuscript, Page 3, rows 105-107) if the Authors could find an appropriate correction of the formula to prevent negative values (according to the definition provided here and in the Authors´ earlier study /Ref. 15/, the result should be between 2 and 38).
I agree with your objection. Many years ago, I considered improving the KEB without negative values. Finally this shortcoming turned into an advantage for routine medical practice. Negative KEB values were generally accepted as a deep anaerobic metabolism associated with an intensive inflammatory response with an oxidative burst of professional phagocytes. Your present objection, however, motivates me to revisit the issue.
- The second reason is that KEB in the present form cannot be calculated in cases with unmeasurably low glucose. Such cases are not uncommon (I suppose value of 0.1 mmol/L was used by the Authors in such cases instead). The Authors might wish to discuss this issue in the Discussion section.
We use as the measurement limit for glucose concentration of 0.11 mmol/L. I have added this information and the way of solution to the “Material and Methods” section.
- Possible relevance of adding pyruvate determination in further studies could be discussed as well; namely pyruvate is the metabolite standing on the intersection of different metabolic pathways as showed in Figure 1.
Many years ago I analyzed pyruvate in the CSF. I derived an equation called "Energy Efficiency" involving pyruvate concentration (Kelbich et al. Evaluations of the energy relations in the CSF compartment by investigation of selected parameters of the glucose metabolism in the CSF. Klin. Biochem. Metab. 1998, 6, 213–225. Kelbich et al. Findings in Cerebrospinal Fluid in a Patient with Bacterial Meningitis – Case Repost. Klin. Biochem. Metab. 2002, 10, 54-68.) Unfortunately, later Roche Company finished production of reagents for our analyzer. Therefore, I had to stop my work. I have not returned to this subject since.
Sincerely yours,
Petr Kelbich et al.
